# 2-Phenylpyridine Derivatives: Synthesis and Insecticidal Activity against *Mythimna separata*, *Aphis craccivora*, and *Tetranychus cinnabarinus*

**DOI:** 10.3390/molecules28041567

**Published:** 2023-02-06

**Authors:** Wenliang Zhang, Jingjing Chen, Xiaohua Du

**Affiliations:** Catalytic Hydrogenation Research Center, Zhejiang Key Laboratory of Green Pesticides and Cleaner Production Technology, Zhejiang Green Pesticide Collaborative Innovation Center, Zhejiang University of Technology, Hangzhou 310014, China

**Keywords:** insecticide synthesis, 2-phenylpyridine, *N*-phenylbenzamide, insecticidal activities

## Abstract

The increase in the insecticide resistance of pests, such as *Mythimna separata*, *Aphis craccivora Koch*, and *Tetranychus cinnabarinus*, necessitates the development of new heterocyclic compounds with high insecticidal activity. A series of novel 2-phenylpyridine derivatives containing *N*-phenylbenzamide moieties were designed and synthesised with Suzuki–Miyaura cross-coupling, nucleophilic substitution, and amidation reactions. The reaction conditions in each step are mild, and the product is easy to separate (yield is about 85%). The structures of the compounds were characterised using ^1^H and ^13^C NMR spectroscopy and HRMS. Moreover, the insecticidal activity of the compounds was analysed using the leaf dipping method. The compounds **5a**, **5d**, **5g**, **5h**, and **5k** at 500 mg/L exhibited 100% inhibition against *Mythimna separata*. Therefore, the 2-phenylpyridine moieties have the potential to lead to the discovery of novel and effective insecticides.

## 1. Introduction

Pyridine is a nitrogen-containing heterocyclic compound. Pyridine derivatives exhibit various biological activities and are generally used to protect crops. Herbicides [1,2], fungicides [3,4,5,6], insecticides [7,8,9], and plant growth regulators [10,11] generally contain pyridine-based compounds. Compounds containing 2-phenylpyridine have exhibited high biological activity [12,13]; hence, herbicides [14], fungicides [15], and insecticides [16] containing 2-phenylpyridine are frequently reported (Figure 1).

Amide compounds are of fundamental importance in pesticide chemistry [17]. Many pesticides, such as insecticides [18,19], fungicides [20], and herbicides [21], contain amide-based compounds (Figure 2). Chlorantraniliprole [22,23], a commercial insecticide created by DuPont, is a new, efficient, and low-toxicity systemic insecticide that not only exhibits high activity against pests but is also safe for users and beneficial for insects. Owing to its excellent efficacy, chlorantraniliprole has been widely used to protect crops, such as soybean, rice, cotton, and corn.

The resistance of pests toward insecticides has increased and considerably affected the growth and harvest of crops [24,25]. Wang et al. demonstrated that flubendiamide exhibited certain insecticidal activity against a series of pests such as lepidoptera pests [26]. A class of chlorantraniliprole analogues was prepared by Ying et al. using pyridyl pyrazolidone carboxylic acid compounds and benzoic acid derivatives as the starting materials, which showed good insecticidal activity [27]. Liu et al. reported that a class of novel phenylpyridine derivatives exhibited excellent insecticidal activity against harmful mites such as *Tetranychus cinnabarinus*, etc., and can obtain very good effects at a very low dose. [16]. Phenylpyridine [13,14] and amide [17,18,19] are important components of pesticides; therefore, we designed a series of new chemicals whose structures were based on these two compounds. To obtain highly active compounds, 11 2-phenylpyridine compounds containing *N*-phenylbenzamide moieties were synthesised according to the substructure link principle. The synthesised compounds were analysed using high-resolution mass spectrometry (HRMS) and nuclear magnetic resonance (NMR) spectroscopy, and their insecticidal activity against *Mythimna separata* (MS.), *Aphis craccivora Koch* (AC.), and *Tetranychus cinnabarinus* was determined.

## 2. Results and Discussion

### 2.1. Chemistry

The compounds were synthesised as illustrated in Figure 1. Compound **3** was synthesised using Suzuki–Miyaura cross-coupling of 2,3-dichloro-5-trifluoromethylpyridine (**1**) and 4-hydroxyphenylboronic acid (**2**), whereas **4** was synthesised using the nucleophilic substitution of **3** with 5-fluoro-2-nitrobenzoic acid. Using **4** and O-(1H-benzotriazol-1-yl)-N,N,N′,N′-tetramethyiuronium hexalfuorophosphate (HBTU) as the reactants and CH_2_Cl_2_ as the solvent, a series of designed compounds were obtained in high yields using an amidation reaction. The target compounds (**5a**–**5k**) were analysed using HRMS and NMR. All spectral and analytical data were consistent with the assigned structures (Appendix A).

### 2.2. Greenhouse Insecticidal Activity Assays

As shown in Table 1, all compounds exhibited high insecticidal activities against *Mythimna separata* but were ineffective against *Tetranychus cinnabarinus*. Among the tested compounds, **5b**, **5d**, **5g**, **5h**, and **5k** displayed 100% inhibitory activities against *Mythimna separata*. However, the rest of the tested compounds showed lower inhibitory activities of ~70% against *Mythimna separata*.

The relationship between the structure and activity of the compounds was investigated, and we found that when the substituent in the second position of the benzene was methoxy or in the third position was chlorine, trifluoromethyl, and trifluoromethoxy, the insecticidal activities of the synthesised compounds were optimal. Further studies on the insecticidal activity of 2-phenylpyridine compounds containing *N*-phenylbenzamide moieties are in progress.

## 3. Materials and Methods

### 3.1. Instrumentation

All reagents and other materials were purchased from commercial sources and used without additional purification unless otherwise noted. A B-545 melting point instrument was used to determine the melting point, without calibration. A Bruker AV-400 or AV-500 MHz spectrometer was used to generate NMR spectra, with CDCl_3_ as the solvent. An Agilent 6545 Q-TOF liquid chromatography-mass spectrometer was used for mass spectrometry.

### 3.2. Synthesis

The synthesis of the compounds is outlined in Figure 2. Compound **3** was synthesised using Suzuki–Miyaura cross-coupling, **4** using nucleophilic substitution, and **5a**–**5k** were synthesised using an amidation reaction.

#### 3.2.1. General Approach to the Synthesis of **3**

To a stirred solution of 2,3-dichloro-5-(trifluoromethyl)pyridine (1.08 g, 5 mmol), 4-hydroxybenzeneboronic acid (0.76 g, 5.5 mmol), potassium carbonate (1.38 g, 10 mmol), and triphenylphosphine (0.13 g, 10 mol%) in acetonitrile (10 mL) and methanol (10 mL), palladium(II) acetate (0.06 g, 5 mol%) was added under a nitrogen atmosphere. The mixture was stirred at 50 °C for 6 h, and water was then slowly added to the reaction solution. The mixture was extracted thrice (30 mL × 3) with ethyl acetate, and the combined organic layers were then washed with brine and dried over MgSO_4_. The solution was concentrated in vacuo and **3** was obtained using recrystallisation in ethanol and water.

#### 3.2.2. General Approach to the Synthesis of **4**

Caesium carbonate (60 mmol) was added to a stirred solution of 4-(3-chloro-5-(trifluoromethyl)pyridin-2-yl) phenol (20 mmol) in *N,N*-dimethylformamide (100 mL). The mixture was stirred for approximately 0.5 h and then 5-fluoro-2-nitrobenzoic acid (21 mmol) was added. The mixture was then stirred at 100 °C for 9 h. Water was then slowly added to the reaction solution to quench the reaction. The mixture was extracted thrice (30 mL × 3) with ethyl acetate. The combined organic layers were then washed with brine and dried over MgSO_4_. The solution was concentrated in vacuo, and **4** was used directly for the next reaction.

#### 3.2.3. General Approach to the Synthesis of **5a**–**5k**

Substituted aniline (2.2 mmol) was added to a stirred solution containing 5-(4-(3-chloro-5-(trifluoromethyl)pyridin-2-yl)phenoxy)-2-nitrobenzoic acid (2 mmol), HBTU (2.2 mmol), and triethylamine (2.2 mmol) in CH_2_Cl_2_ (15 mL). The mixture was stirred at 25 °C for 8 h. A saturated potassium bisulphate solution was then added to the reaction solution, and the mixture was extracted thrice (30 mL × 3) with ethyl acetate. The combined organic layers were then washed with brine and dried over MgSO_4_. The solution was concentrated in vacuo and then subjected to silica gel column chromatography using petroleum ether/ethyl acetate (7:1, *v/v*) to obtain the target compounds (**5a**–**5k**).

Description of 5-(4-(3-chloro-5-(trifluoromethyl)pyridin-2-yl)phenoxy)-2-nitro-N-phenylbenzamide (**5a**): White solid, yield 76.5%. m.p. 193.2–195.1 °C; ^1^H NMR (500 MHz, CDCl_3_) δ 8.85 (s, 1H), 8.11 (d, J = 9.1 Hz, 1H), 8.08 (d, J = 1.5 Hz, 1H), 7.92 (s, 1H), 7.87 (d, J = 8.6 Hz, 2H), 7.54 (d, J = 7.8 Hz, 2H), 7.34 (t, J = 7.8 Hz, 2H), 7.22 (d, J = 8.6 Hz, 2H), 7.16 (dd, J = 7.0, 3.8 Hz, 2H), 7.12 (dd, J = 9.1, 2.6 Hz, 1H); ^13^C NMR (101 MHz, CDCl_3_) 162.05, 155.47, 144.38, 137.14, 135.55, 131.91, 130.19, 129.21, 127.50, 125.30, 120.51, 120.14, 118.54, 117.00. HRMS calcd. for C_25_H_14_ClF_3_N_3_O_4_ [M–H]^−^ 512.0630, found 512.0628.

Description of 5-(4-(3-chloro-5-(trifluoromethyl)pyridin-2-yl)phenoxy)-N-(3-chlorophenyl)-2-nitrobenzamide (**5b**): White solid, yield 73.8%. m.p. 209.4–211.7 °C; ^1^H NMR (500 MHz, CDCl_3_) δ 8.77 (d, J = 1.1 Hz, 1H), 8.09 (d, J = 9.3 Hz, 1H), 8.00 (d, J = 1.6 Hz, 1H), 7.81 (d, J = 8.6 Hz, 2H), 7.60 (d, J = 6.5 Hz, 2H), 7.34 (d, J = 7.6 Hz, 1H), 7.21 (d, J = 8.0 Hz, 1H), 7.16 (s, 1H), 7.14 (s, 1H), 7.10 (s, 1H), 7.09–7.08 (m, 1H), 7.08–7.06 (m, 1H); ^13^C NMR (101 MHz, CDCl_3_) δ 163.96, 162.15, 158.52, 155.37, 144.37, 144.33, 144.29, 140.14, 138.28, 135.61, 135.57, 135.11, 134.84, 134.45, 131.92, 130.17, 127.53, 125.32, 120.56, 120.18, 118.66, 118.41, 116.86. HRMS calcd. for C_25_H_13_Cl_2_F_3_N_3_O_4_ [M–H]^−^ 546.0241, found 5446.0239.

Description of 5-(4-(3-chloro-5-(trifluoromethyl)pyridin-2-yl)phenoxy)-N-(2,4-dimethylphenyl)-2-nitrobenzamide (**5c**): White solid, yield 77.6%. m.p. 175.1–177.2 °C; ^1^H NMR (500 MHz, CDCl_3_) δ 8.77 (s, 1H), 8.04 (t, J = 8.7 Hz, 1H), 7.98 (dd, J = 9.5, 1.4 Hz, 1H), 7.83–7.74 (m, 2H), 7.51 (d, J = 7.9 Hz, 1H), 7.28 (d, J = 17.3 Hz, 1H), 7.15 (d, J = 5.7 Hz, 1H), 7.12 (dd, J = 14.7, 6.8 Hz, 2H), 7.03 (dd, J = 9.0, 2.6 Hz, 1H), 6.95 (d, J = 10.8 Hz, 2H), 2.22 (s, 3H), 2.15 (s, 3H); ^13^C NMR (101 MHz, CDCl_3_) δ 164.31, 161.93, 158.57, 155.56, 144.34, 140.45, 136.20, 135.80, 135.58, 135.55, 134.31, 131.94, 131.87, 131.33, 130.74, 130.19, 127.45, 126.19, 124.43, 124.01, 120.06, 118.27, 117.45, 21.29, 17.65. HRMS calcd. for C_27_H_18_ClF_3_N_3_O_4_ [M–H]^−^ 540.0943, found 540.0946.

Description of 5-(4-(3-chloro-5-(trifluoromethyl)pyridin-2-yl)phenoxy)-N-(2-methoxyphenyl)-2-nitrobenzamide (**5d**): Yellow solid, yield 80.3%. m.p. 151.3–154.2 ℃; ^1^H NMR (500 MHz, CDCl_3_) δ 8.86 (d, J = 0.9 Hz, 1H), 8.45 (dd, J = 7.9, 1.2 Hz, 1H), 8.18 (d, J = 9.1 Hz, 1H), 8.08–8.03 (m, 2H), 7.89 (d, J = 8.6 Hz, 2H), 7.25 (s, 1H), 7.24–7.21 (m, 2H), 7.16 (dd, J = 9.1, 2.7 Hz, 1H), 7.13–7.10 (m, 1H), 7.02 (t, J = 7.4 Hz, 1H), 6.91 (d, J = 7.7 Hz, 1H), 3.86 (s, 3H); ^13^C NMR (101 MHz, CDCl_3_) δ 163.40, 161.90, 158.56, 155.56, 148.01, 144.38, 144.34, 140.55, 135.76, 135.58, 134.31, 131.88, 130.17, 127.47, 127.17, 126.51, 124.66, 121.27, 120.22, 120.09, 118.41, 117.18, 110.08, 55.73. HRMS calcd. for C_26_H_16_ClF_3_N_3_O_5_ [M–H]^−^ 542.0736, found 542.0741.

Description of N-(3-chloro-2-methylphenyl)-5-(4-(3-chloro-5-(trifluoromethyl)pyridin-2-yl)phenoxy)-2-nitrobenzamide (**5e**): White solid, yield 81.9%. m.p. 172.3–174.6 °C; ^1^H NMR (500 MHz, CDCl_3_) δ 8.86 (s, 1H), 8.15 (t, J = 8.6 Hz, 1H), 8.08 (d, J = 1.6 Hz, 1H), 7.88 (t, J = 7.6 Hz, 2H), 7.65 (d, J = 7.9 Hz, 1H), 7.53 (d, J = 12.6 Hz, 1H), 7.27 (s, 1H), 7.24 (d, J = 8.4 Hz, 2H), 7.21 (d, J = 2.0 Hz, 1H), 7.18 (t, J = 7.2 Hz, 1H), 7.14 (dd, J = 9.2, 2.4 Hz, 1H), 2.31 (s, 3H); ^13^C NMR 101 MHz, CDCl_3_) δ 164.36, 162.11, 158.52, 155.42, 144.36, 140.31, 135.71, 135.60, 135.56, 135.31, 135.09, 134.83, 134.45, 131.91, 130.20, 129.49, 127.58, 127.41, 127.16, 123.28, 120.11, 118.43, 117.25, 14.74. HRMS calcd. for C_26_H_15_Cl_2_F_3_N_3_O_4_ [M–H]^−^ 560.0397, found 560.0399.

Description of N-(4-chloro-2-methylphenyl)-5-(4-(3-chloro-5-(trifluoromethyl)pyridin-2-yl)phenoxy)-2-nitrobenzamide (**5f**): White solid, yield 80.2%. m.p. 199.2–201.7 °C; ^1^H NMR (400 MHz, CDCl_3_) δ 8.88 (s, 1H), 8.21 (d, J = 9.1 Hz, 1H), 8.10 (s, 1H), 7.91 (d, J = 8.4 Hz, 2H), 7.82 (d, J = 8.3 Hz, 1H), 7.28–7.16 (m, 7H), 2.29 (s, 3H); ^13^C NMR (101 MHz, CDCl_3_) δ 164.20, 162.10, 158.51, 155.44, 144.41, 144.37, 140.33, 135.60, 135.57, 135.40, 134.45, 133.24, 132.08, 131.91, 131.41, 130.48, 130.19, 127.58, 126.98, 125.35, 120.12, 118.47, 117.56, 117.23, 17.92. HRMS calcd. for C_26_H_15_Cl_2_F_3_N_3_O_4_ [M–H]^−^ 560.0397, found 560.0402.

Description of N-(5-chloro-2-methylphenyl)-5-(4-(3-chloro-5-(trifluoromethyl)pyridin-2-yl)phenoxy)-2-nitrobenzamide (**5g**): white solid, yield 74.9%. m.p. 184.2–186.4 °C; ^1^H NMR (400 MHz, CDCl_3_) δ 8.88 (s, 1H), 8.21 (d, J = 9.1 Hz, 1H), 8.10 (s, 1H), 7.91 (d, J = 8.4 Hz, 2H), 7.82 (d, J = 8.3 Hz, 1H), 7.28–7.16 (m, 7H), 2.29 (s, 3H); ^13^C NMR (101 MHz, CDCl_3_) δ 164.10, 162.14, 158.53, 155.41, 144.39, 140.25, 135.68, 135.60, 135.27, 134.46, 132.25, 131.92, 131.51, 130.20, 127.98, 127.56, 126.57, 126.24, 126.08, 123.61, 120.15, 118.49, 117.18, 17.32. HRMS calcd. for C_26_H_15_Cl_2_F_3_N_3_O_4_ [M–H]^−^ 560.0397, found 560.0404.

Description of 5-(4-(3-chloro-5-(trifluoromethyl)pyridin-2-yl)phenoxy)-2-nitro-N-(3-(trifluoromethoxy)phenyl)benzamide (**5h**): brown solid, yield 76.1%. m.p. 140.2–141.4 °C; ^1^H NMR (500 MHz, CDCl_3_) δ 8.85 (s, 1H), 8.12 (t, J = 7.6 Hz, 1H), 8.08 (d, J = 1.6 Hz, 1H), 7.87 (d, J = 7.0 Hz, 2H), 7.85 (d, J = 7.1 Hz, 1H), 7.46–7.42 (m, 4H), 7.22 (d, J = 8.6 Hz, 2H), 7.13 (dd, J = 8.4, 5.8 Hz, 2H); ^13^C NMR (101 MHz, CDCl_3_) δ 185.24, 164.30, 162.18, 158.52, 155.31, 144.34, 144.30, 139.95, 137.74, 135.62, 135.58, 134.93, 134.43, 131.90, 131.65, 130.23, 129.75, 127.45, 123.57, 121.76, 120.19, 118.61, 117.14, 116.82. HRMS calcd. for C_26_H_13_ClF_6_N_3_O_5_ [M–H]^−^ 596.0415, found 596.0410.

Description of 5-(4-(3-chloro-5-(trifluoromethyl)pyridin-2-yl)phenoxy)-N-(2,4-difluorophenyl)-2-nitrobenzamide (**5i**): White solid, yield 78.1%. m.p. 159.1–161.5 °C; ^1^H NMR (400 MHz, CDCl_3_) δ 8.84 (s, 1H), 8.25 (s, 1H), 8.09 (dd, J = 5.1, 3.7 Hz, 2H), 7.86 (d, J = 8.6 Hz, 2H), 7.52 (s, 1H), 7.42 (d, J = 8.1 Hz, 1H), 7.33 (t, J = 8.2 Hz, 1H), 7.22 (d, J = 8.5 Hz, 2H), 7.15 –7.10 (m, 2H), 7.02 (d, J = 8.1 Hz, 1H); ^13^C NMR (101 MHz, CDCl_3_) δ 163.78, 162.18, 158.52, 155.36, 144.36, 140.24, 135.60, 134.94, 134.50, 131.93, 130.20, 127.62, 126.25, 123.49, 123.38, 120.21, 118.70, 116.89, 111.70, 111.48, 103.80. HRMS calcd. for C_25_H_13_ClF_5_N_3_O_4_ [M–H]^−^ 548.0422, found 548.0422.

Description of 5-(4-(3-chloro-5-(trifluoromethyl)pyridin-2-yl)phenoxy)-2-nitro-N-(3-(trifluoromethyl)phenyl)benzamide (**5j**): white solid, yield 74.3%. m.p. 146.1–148.7 °C; ^1^H NMR (400 MHz, CDCl_3_) δ 8.88 (s, 1H), 8.15 (d, J = 8.5 Hz, 1H), 8.10 (s, 1H), 7.89 (d, J = 8.7 Hz, 2H), 7.34 (d, J = 8.5 Hz, 2H), 7.22 (d, J = 8.7 Hz, 2H), 7.13 (d, J = 8.5 Hz, 2H), 6.91 (d, J = 8.6 Hz, 2H), 4.59 (d, J = 5.4 Hz, 2H), 3.82 (s, 3H); ^13^C NMR (101 MHz, CDCl_3_) δ 164.24, 162.15, 158.53, 155.33, 149.53, 144.33, 144.29, 139.91, 138.62, 135.61, 135.58, 134.95, 134.39, 131.88, 130.22, 127.40, 126.57, 126.23, 120.16, 118.57, 118.44, 117.16, 116.83, 113.20. HRMS calcd. for C_26_H_13_ClF_6_N_3_O_4_ [M–H]^−^ 596.0453, found 596.0447.

Description of N-(4-acetylphenyl)-5-(4-(3-chloro-5-(trifluoromethyl)pyridin-2-yl)phenoxy)-2-nitrobenzamide (**5k**): white solid, yield 77.6%. m.p. 196.1–198.3 °C; ^1^H NMR (400 MHz, CDCl_3_) δ 8.86 (s, 1H), 8.19 (d, J = 8.9 Hz, 1H), 8.09 (s, 1H), 8.04 (s, 1H), 7.97 (d, J = 8.5 Hz, 2H), 7.89 (d, J = 8.5 Hz, 2H), 7.69 (s, 1H), 7.25–7.16 (m, 5H), 2.60 (s, 3H); ^13^C NMR (101 MHz, CDCl_3_) δ 164.01, 162.17, 158.49, 155.34, 144.38, 144.34, 144.31, 141.60, 140.13, 135.58, 135.07, 134.49, 133.59, 131.93, 130.20, 129.82, 127.55, 120.16, 119.55, 118.73, 116.84, 29.71, 26.52. HRMS calcd. for C_27_H_16_ClF_3_N_3_O_5_ [M–H]^−^ 554.0736, found 554.0737.

### 3.3. Insecticidal Activity Test

The insecticidal activities of **5a**–**5k** against *Mythimna separata*, *Aphis craccivora Koch*, and *Tetranychus cinnabarinus* were evaluated according to a previously reported method [28,29,30]. First, all test compounds were dissolved in *N,N*-dimethylformamide containing 0.1% Tween-80 emulsifier and were then diluted to the corresponding dosage with distilled water. Appropriate amounts of corn leaves were fully soaked in liquid medicine for 10 s, dried naturally in the shade, and placed in a culture dish with filter paper. The larvae of *Mythimna separata, Aphis craccivora Koch*, and *Tetranychus cinnabarinus* were placed indoors on the processed corn leaves at 24–27 °C. The results were obtained after three days; the insect was touched with a brush, and it was regarded dead if there was no response. The control check (CK) group used only distilled water without any compounds, and the results of insecticidal activities are reported in Table 1.

## 4. Conclusions

A series of new 2-phenylpyridine derivatives containing *N*-phenylbenzamide moieties were designed and synthesised as possible insecticidal candidates. Compounds **5b**, **5d**, **5g**, **5h**, and **5k** exhibited high insecticidal activities (100%) against *Mythimna separata* at 500 mg/L. Thus, the results of this study will provide a novel direction for the further use of 2-phenylpyridine derivatives as insecticides.

## Data Availability

Not applicable.

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
