# Peer review of "2-Phenylpyridine Derivatives: Synthesis and Insecticidal Activity against Mythimna separata, Aphis craccivora, and Tetranychus cinnabarinus"

_molecules, 2023, doi:10.3390/molecules28041567_

Round 1
Reviewer 1 Report
see attached

Author Response
Thank you very much for your kind consideration of this manuscript. We have modified it as required. Please see the attachment.

Reviewer 2 Report
Comments
Title
· The title of this manuscript should be changed to “Synthesis and insecticidal activity of novel 2-phenylpyridine derivatives against the armyworm, Mythimna separata, the cowpea aphid, Aphis craccivora, and the carmine spider mite, Tetranychus cinnabarinus”.
Abstract:
· Overall: More details are needed here.
· Please give a background on why these particular insects were selected.
· Methods used for insecticidal activity evaluation not provided.
· Additionally, no sufficient details on either synthesis or insecticidal activity results are found here.
Introduction
· A background paragraph on the economic importance of the model insects with some recent references on related compounds used in their control is needed.
· Line 38: “Based on the upper study” which one?
· Aphiscraccivora Koch (AK.): replace with Aphis craccivora Koch (AC.) in the whole manuscript.
Results and Discussion
· Identify the following: HBTU (line 55); HRMS and NMR (Line 57) and Ck in table 1.
· Line 64: Replace mythimna separata with Mythimna separata.
Table 1
· Since this table contains the main findings of insecticidal activity against the three tested insect pests, these are some questions need be answered.
1. Why only (500 mg/L) dosage was selected. A range of serial concentrations is recommended in such studies.
2. To the best of my knowledge, this dosage is too high to be applied for pest control.
3. Did you consider the effect of tested dosage on the non-target organisms?
4. What does (CK) mean here?
5. Did you used a positive control? From the point of view of the biological assay, negative and positive controls were absent or not sufficiently described, compromising the statistical magnitude of the results presented. Insert negative and positive controls into biological assays.
6. For 5k compound, does the inhibitory percent (900) is correct for Mythimna separata.
Materials and Methods
· Identify: LCMS (Line 80); DMF (Line 201);
· Line 79: Replace CDCl3 with CdCl3.
· Line 81: Synthesis of test compounds needs more details here
· Line 199: Replace “mythimna separata (M. s.), aphiscraccivora Koch (A. k.), and tetranychus cinnabarinus (T. c.)” with “Mythimna separata, aphis craccivora and tetranychus cinnabarinus”.
· Line 200: “ … were evaluated in the light of the previously reported method [25-27]” these reported methods doesn’t explain well the used methods for insecticidal activity evaluation, please consider revision then give a brief on the used method even after cite the proper references.
· Lines 199-209: This paragraph needs to be re-written; language is too hard to be understood.
Overall
· The language of this article needs to be revised by a native speaker.

Author Response

(The authors gave the same response as above.)

Round 2
Reviewer 1 Report
The main problem with the paper is that the English remains very poor. The authors don’t seem to understand basic sentence construction. For example, the first sentence of the Introduction (line 27):
Pyridine is a heterocycle nitrogen-containing. The adjective (nitrogen-containing) should be before the noun (heterocycle) not after. The sentence should read: Pyridine is a nitrogen-containing heterocycle. Better still: Pyridine is a nitrogen-containing heterocyclic compound.
The second sentence (lines 30-31): Appearing in the structure of herbicides [1,2], fungicides [3-6], insecticides [7-9] and plant growth regulators [10,11]. Here, this is not a proper sentence as there is no subject (what is appearing in the structure of… ?). A simple sentence must contain a subject-verb-object in that order. Presumably the authors mean: The scaffold pyridine occurs in the structure of herbicides…. The authors need to find a native English speaker to correct the English throughout the paper.
I said in my previous report that it is very difficult to assess just how active these herbicides are as there is no comparison with a known insecticide. The authors should include the activity for a known insecticide (such as chlorantraniliprole). They have not done this. The blank (control check) with distilled water doesn’t give any useful information.
Other corrections:
In Scheme 1, there are no structure numbers in the first line, so that these structures can’t be referred to in the text. Also, the first arrow is a reaction arrow, so that as drawn, it looks as if the two compounds with sections highlighted in red are reacting together to give a mixed structure (clearly an impossible reaction). The authors need to clarify this.
In conclusion, the English is still not good enough for publication in Molecules.
Author Response
Dear Reviewer,
We have revised the manuscript. Please see the attachment.
Kind regards,
Wenliang Zhang

Reviewer 2 Report
Comments R2
Abstract:
· Please revise the inserted sentence on model organisms.
· Still, no sufficient details on either synthesis or insecticidal activity results are found here.
Introduction
· A background paragraph on the economic importance of the model insects with some recent references on related compounds used in their control is needed. Not yet addressed.
Table 1
· Since this table contains the main findings of insecticidal activity against the three tested insect pests, these are some questions NOT answered.
1. Did you consider the effect of tested dosage on the non-target organisms?
2. Did you used a positive control? From the point of view of the biological assay, negative and positive controls were absent or not sufficiently described, compromising the statistical magnitude of the results presented. Insert negative and positive controls into biological assays.
· Finally, the language still needs to be revised.
Author Response

(The authors gave the same response as above.)
